# Therapeutic Implications of TGF-β Pathway in Desmoid Tumor Based on Comprehensive Molecular Profiling and Clinicopathological Properties

**DOI:** 10.3390/cancers14235975

**Published:** 2022-12-02

**Authors:** Kum-Hee Yun, Changhee Park, Hyang Joo Ryu, Chan-Young Ock, Young Han Lee, Wooyeol Baek, Hong In Yoon, Yoon Dae Han, Sang Kyum Kim, JooHee Lee, Seong-Jin Kim, Kyung-Min Yang, Seung Hyun Kim, Hyo Song Kim

**Affiliations:** 1Department of Internal Medicine, Graduate School of Medical Science, Brain Korea 21 Project, Yonsei University College of Medicine, Seoul 03722, Republic of Korea; 2Department of Internal Medicine, Seoul National University Hospital, Seoul 03722, Republic of Korea; 3Department of Pathology, Severance Hospital, Yonsei University College of Medicine, Seoul 03722, Republic of Korea; 4Bang & Ock Consulting Inc., Seoul 03722, Republic of Korea; 5Department of Radiology, Yonsei University College of Medicine, Seoul 03722, Republic of Korea; 6Department of Plastic Surgery, Yonsei University College of Medicine, Seoul 03722, Republic of Korea; 7Department of Radiation Oncology, Yonsei University College of Medicine, Seoul 03722, Republic of Korea; 8Department of Surgery, Yonsei University College of Medicine, Seoul 03722, Republic of Korea; 9Medpacto, Inc., Seoul 03722, Republic of Korea; 10Department of Orthopedic Surgery, Yonsei University College of Medicine, Seoul 03722, Republic of Korea; 11Department of Internal Medicine, Yonsei University College of Medicine, Seoul 03722, Republic of Korea

**Keywords:** desmoid tumor, rare cancer, next-generation sequencing, TGF-β pathway, combination treatment, patient-derived primary cell model

## Abstract

**Simple Summary:**

Desmoid tumors are rare and aggressive tumors that currently do not receive satisfactory systemic treatment. Thus, an in-depth study into desmoid tumors was needed. In this study, we performed the comprehensive molecular profiling for desmoid tumor samples from patients and found the druggable biomarker. Notably, the TGF-β signaling pathway was consistently identified as enriched in desmoid tumors using comprehensive molecular profiling. It suggested that therapeutic interventions targeting TGF-β in fibroblasts using TGF-β receptor inhibitors may be clinically beneficial in patients with desmoid tumors. After that, we validated it by using a desmoid patient-derived primary cell model that displayed high concordance with desmoid primary tissues. Finally, we proved that that the inhibition of the TGF-β pathway is useful as a potential treatment for patients with desmoid tumors.

**Abstract:**

(1) Background: Desmoid tumors have a relatively high local failure rate after primary treatment using surgery and/or radiotherapy. Moreover, desmoid tumors recur at the primary site for many patients. An effective therapeutic strategy for the desmoid tumor is needed to maintain quality of life and prolong survival. (2) Method: First of all, we collected desmoid tumor tissues and investigated the status of protein expression for beta-catenin and alpha-SMA through immunohistochemistry. Then, we performed targeted sequencing and whole RNA sequencing. To compare the data with other cancer types, we used NGS data from sarcoma patients at Yonsei Cancer Center (YCC-sarcoma cohort, *n* = 48) and The Cancer Genome Atlas (TCGA, *n* = 9235). Secondly, we established the novel patient-derived preclinical models (*n* = 2) for the validation of treatment strategy. The same gene alteration of primary tissue was demonstrated. (3) Results: We discovered specific gene sets related to the TGF-β signaling pathway. Moreover, we selected the combination treatment comprising TGF-β inhibitor, vactosertib, and imatinib. In screening for the anti-proliferation effect, the combination treatment of TGF-β inhibitor was more effective for tumor suppression than monotherapy. (4) Conclusion: We found preclinical indications that TGF-β inhibitors could prove useful as a potential treatment for patients with desmoid tumors. Moreover, we could find some examples in clinical trials.

## 1. Introduction

Desmoid tumors (also known as aggressive fibromatosis) represent a type of aggressive borderline tumor derived from connective tissues over the whole body. Nowadays, many patients are offered a watchful waiting approach after diagnosis, because DTs sometimes have the possibility of stabilization and spontaneous regression [1]. When the desmoid starts to grow, major treatment for primary care is surgery [2,3]. However, most patients with desmoid tumor have difficulty with recurrence after surgery [4,5], with some spending their whole life in pain. Because of this, many clinical trials for chemotherapy in the desmoid tumor have been carried out, including hormonal blockade, cytotoxic chemotherapy, and tyrosine kinase inhibitors, such as imatinib, sunitinib, and sorafenib, which have achieved partial success with a 5–25% response rate [6,7,8]. Recently, in a single-center phase 2 study, nirogacestat, a gamma-secretase inhibitor, achieved 29% partial response. Moreover, the gamma-secretase inhibitor AL102 is ongoing on a phase 2/3, randomized, multicenter study to evaluate AL102 in patients with progressing desmoid tumors. As a result of these trials, the response rate was improved. Nevertheless, we do not know the mechanism of action for the drug exactly. For in-depth research into desmoid tumors, systemic molecular biological understanding is needed.

With the significant advancement of next-generation-sequencing (NGS), desmoid tumors have been classified into two types by specific genetic characteristics: Sporadic desmoid and familial adenomatous polyposis (FAP). The sporadic desmoid tumor generally has the somatic mutation in CTNNB1, making a protein called beta-catenin. It had high dominancy (85%) in all desmoid patients. FAP is associated with germline APC mutation [9]. Although the molecular pathogenesis of desmoid tumors is largely understood, clinically druggable targets have been not discovered.

Conventional cell lines are convenient and easy to access but correspond poorly with pre-clinical models and clinical results [10,11]. Therefore, it is important to establish preclinical models that closely resemble actual tumors. Owing to the rarity of desmoid tumors, it is particularly important to establish novel preclinical models that reflect their clinical behavior and molecular profiles for drug development.

In this study, first, we collected desmoid primary tissues and conducted comprehensive molecular profiling analyses of desmoid tumors using targeted sequencing (target-seq) and RNA sequencing (RNA-seq) data, then fused these data with the clinicopathological data. Through these process and analysis, we identified the potential druggable biomarkers. Secondly, we established the patient-derived primary cell (PDC) and analyzed it by using NGS tool (target-seq). Moreover, we obtained the highly concordant PDC with the genetic alteration of desmoid tissue. Finally, we discovered therapeutic implication for dependency on the TGF- β pathway of desmoid tumor. Additionally, we suggested the combination treatment with TGF- β receptor inhibitor.

## 2. Materials and Methods

### 2.1. Patients and Treatment

The study, approved by the institutional review board (IRB No.4-2019-0804), included 33 patients with desmoid tumors and 51 patients with sarcoma. Surgically resected formalin-fixed paraffin-embedded (FFPE) tissue samples obtained prior to radiotherapy or chemotherapy were used to target-seq and RNA-seq. All patients provided written informed consent before enrollment in the study in accordance with the Declaration of Helsinki and Guidelines for Good Clinical Practice.

### 2.2. Tumor Sample Collection and Genomic Analysis

Tumor tissues were obtained prior to the initiation of the study. If the tumor content was estimated to be ≥40% after pathological review, DNA and RNA were extracted from the obtained tissues using a GeneRead DNA FFPE Kit (Qiagen, Hilden, Germany). Total RNA was extracted and purified from frozen tumor samples using the ReliaPrep FFPE Total RNA Miniprep System (Promega).

### 2.3. DNA Sequence Generation and Alignment

The quality and quantity of purified DNA were assessed using fluorometry (Qubit, Invitrogen, Waltham, MA, USA) and gel electrophoresis. Briefly, 500 ng of genomic DNA from each sample was fragmented by acoustic shearing using a Covaris S2 instrument. Fragments of 150–200 bp were ligated to Illumina adapters and amplified using PCR. The samples were concentrated to 750 ng in 3.4 μL deionized water using a Speedvac machine (Thermo Scientific, Waltham, MA, USA) and hybridized using RNA probes (SureSelect XT Custom 1 kb up to 499 kb (3264191, 1.667 Mbp)) for 16–24 h at 65 °C. The captured targets were pulled down using biotinylated probe/target hybrid streptavidin-coated magnetic beads (Dynabeads My One Streptavidine T1; Life Technologies, Carlsbad, CA, USA). The selected regions were then amplified using Illumina PCR primers and libraries were identified using an Agilent TapeStation 4200 with High Sensitivity D 1000 ScreenTape (Agilent, Santa Clara, CA, USA) and a KAPA Library Quantification Kit (Kapa Biosystems, Potters Bar, UK). High-quality libraries were pooled and sequenced using the Illumina NovaSeq6000 platform (Illumina, San Diego, CA, USA) with 150 bp paired-end reads according to the manufacturer’s protocols. Images were analyzed using NovaSeq6000 control software version 1.3.1 and the output base calling data were de-multiplexed using bcl2fastq version v2.20.0.422 to generate fastQC files.

### 2.4. RNA Sequence Generation and Alignment

The quality and quantity of the purified RNA were assessed using a BioAnalyzer2100 instrument (Agilent) according to the manufacturer’s instructions. The sequencing analysis pipeline is available in the online supplementary methods. RNA libraries were constructed using SMARTer Stranded Total RNA-Seq Kit v2 (Pico Input Mammalian kit, Illumina) with at least 1 ng of RNA per sample. rRNA was removed using a Removal mix and cDNA was synthesized immediately. After 3′-end adenylation and adapter ligation, PCR amplification was performed for 15 cycles, and samples were purified using AMPured beads. Fragment size (250–350 bp) and concentration (>5 ng/μL) were verified using a TapeStation 4200 instrument (Agilent) with D1000 screen tape. Each sample was quantified using a KAPA Library Quantification Kit.

High-quality libraries were pooled and sequenced using the Illumina NovaSeq6000 platform (Illumina) with 150 bp paired-end reads according to the manufacturer’s protocols. Images were analyzed using NovaSeq6000 control software version 1.3.1 and the output base calling data were de-multiplexed using bcl2fastq version v2.20.0.422 to generate fastQC files.

RNA-seq data were generated for 30 patients with desmoid tumors. Adapter sequences were removed using Cutadpt [3]. Quality control check at pre-alignment step was conducted using FASTQC and at post-alignment step using RSeQC [6]. QC results were visualized with MultiQC [7]. During the post-alignment step, we noticed three patients with potential problems in read distribution and inferred experimental criteria. Therefore, they were excluded from further analysis.

RNA-seq data from The Cancer Genome Atlas (TCGA) were obtained from UCSC Xena [12] and corresponding clinical data were obtained from cBioportal [13,14]. The RNA-seq data for the desmoid tumor samples were normalized to those of other sarcomas and TCGA samples using the trimmed mean of M values algorithm [15].

### 2.5. Bioinformatics Analyses

After variant allele frequency data were obtained, silent mutations were filtered, and alterations with potential bias (including clustered events) and strand bias were excluded. Mutations classed as “disease causing” by MutationTaster2 were included to analyze the genomic profiles of desmoid tumors [16]. The tumor mutation burden (TMB) of desmoid tumors and other sarcomas was calculated using the number of synonymous and non-synonymous variants with a variant allele frequency ≥5% divided by the coverage size (megabases) of the sequencing targets (1.134822 megabases for desmoid tumors, 28.393845 megabases for other sarcomas). The TMBs of TCGA data were retrieved from the literature [17].

To cluster the transcriptomic data, we first calculated *p* values using the Student’s *t*-test and fold changes to compare the gene expression values of desmoid tumors and other sarcomas. We selected genes with *p* < $1.0\times 10^{-8}$ and fold changes > 1.33 or < 0.75. Average linkage hierarchical clustering was performed using the Cluster 3.0 software [18]. The clustering results were visualized using Java TreeView 3.0 [19]. The branch of the tree containing genes that were upregulated in desmoid tumors was used for gene ontology and Kyoto Encyclopedia of Genes and Genomes Pathway Analysis using the Database for Annotation, Visualization, and Integrated Discovery (version 6.8) [20,21]. Ten pathways with the most significant *p*-values were selected.

Gene set enrichment analysis (GSEA; version 4.1) was performed, and the top ten pathways with the highest normalized enrichment scores (*p* < 0.05 were selected [22,23]. To create the input data for GSEA, *p*-values were calculated using the Wilcoxon rank-sum test, and fold changes were calculated by comparing gene expression values between desmoid tumors and other sarcomas. Genes were ranked according to the following formula:$$\left(Fold\;change\right)\times -\log_{2}\left(p\;value\right)$$

When analyzing the enriched pathways in desmoid tumors compared with other sarcomas, we used hallmark gene sets provided by the Molecular Signature Database [22,24]. To evaluate the response to TGF-β signaling, we used previously published TGF-β response signatures (TBRS) [25] consisting of four different signatures according to the cells of interest: fibroblasts (F-TBRS), T cells (T-TBRS), macrophages (Ma-TBRS), and endothelial cells (End-TBRS). The list of genes for each TBRS was subjected to GSEA to evaluate TBRS enrichment in desmoid tumors compared with other sarcomas. In addition, the TBRS scores were defined as the sum of the genes for each TBRS.

In the bioinformatic analyses, the statistical details, including the value of *n* and what *n* represents, are available throughout the manuscript or the figures and figure legends. The Wilcoxon rank-sum test was used to compare the values between groups unless otherwise specified, and the data are shown as median values and upper and lower quantile values. Statistical significance was set than 0.05. All statistical analyses were performed using the R software version 4.1.0.

### 2.6. Immunohistochemistry (IHC), Image Acquisition, and Analysis

This method has been described in a previous study [1]. Briefly, the tissues were cut into4-μm-thick sections and subject to heat-induced epitope retrieval. Tissues were subsequently stained with antibodies to β-catenin and alpha-SMA. In β-catenin staining, the subcellular staining pattern. cytoplasmic, membranous, or Golgi staining was regarded as negative expression and nuclear staining was regarded as a positive expression if observed in >1% of desmoid tumor cells. In alpha-SMA staining, when the cytoplasmic expression was present in > 1% of the tumor area, it was regarded as positive. The IHC staining was defined as regardless of the staining intensity for both β-catenin and alpha-SMA.

### 2.7. Establishment of Patient-Derived Preclinical Model

Desmoid tumor samples were surgically removed from the patients who provided informed consent. The specimens were cut into small pieces (2–4 mm) and enzymatically dissociated. The cells were cultured in Minimum Essential medium (Cytiva, Marlborough, UT, USA) supplemented with 10% fetal bovine serum (Gibco, Waltham, MA, USA) and plasmocin prophylactic (Invitrogen, Waltham, MA, USA). The patient-derived cells were maintained at 37 ℃ in a humidified 5% CO_2_ incubator until they reached 80–90% confluence.

### 2.8. Viability and Live/Dead Assay

Each PDC line was seeded in a 96-well plate at a density of 3000 cells/well. After overnight incubation, the cells were starved to stimulate TGF-β and then TGF-β (0.1 ng/mL) was added with vactosertib (MedPacto, Seoul, Korea) and/or imatinib (Selleckchem, Houston, TX, USA) at various concentrations for 96 h. Viability was assessed using WST-8 (BIOMAX, Seoul, Korea) according to the manufacturer’s instructions. The data were visualized and analyzed using GraphPad Prism version 9 (GraphPad Software, San Diego, CA, USA). For the live-dead assay, cells attached to a glass slide were treated as described above and stained using the LIVE/DEAD™ Viability/Cytotoxicity Kit (Invitrogen™).

In the PDC viability analyses, the results were expressed as the mean with standard deviation. Paired Student’s t-tests were used to calculate *p* values. Statistical values are **p* < 0.05, ***p* < 0.01, *** *p* < 0.001, and **** *p* < 0.0001, unless otherwise specified. All statistical analyses were performed using GraphPad Prism software.

### 2.9. In-Cell Western Blotting

The cells were cultured in black, clear-bottom 96-well plates. After treatment with different drugs (Vac; vactosertib monotherapy, IM; imatinib monotherapy, and Vac + IM; combination of vactosertib and imatinib), the cells were fixed with 3.7% formaldehyde solution, washed with 1.0 % Triton X-100 solution in phosphate-buffered saline five times, and blocked using a blocking buffer (Li-COR). The fixed cells were then stained with primary and secondary antibodies using CellTag^TM^ 700 Stain Kits and IRDye 800CW secondary antibodies (Li-COR) according to the manufacturer’s instructions.

## 3. Results

### 3.1. Patient Cohorts

We obtained the available desmoid tumor tissue between January 2019 and January 2022 (Appendix A). The desmoid cohort information is shown in Appendix A. The majority of patients received surgical treatment (90%). Seven patients (23.3%) had a history of chemotherapy with imatinib or vinblastine/MTX. Further, four patients (13.3%) had radiotherapy history. However, desmoid tumors recurred in 16 patients (53.3%) in January 2022.

For validation purposes, through comparison of bioinformatic analyses, we also formed the sarcoma cohort (*n* = 48). The patients in sarcoma cohort were diagnosed during the same period as the desmoid cohort. The sarcoma cohort information is provided in Appendix A.

### 3.2. Patient Clinical and Pathological Characteristics and Genetic Alteration of Desmoid Tissue

Figure 1A shows the clinico-pathological and genetic landscape. The primary site of desmoid tumors was at extra-abdominal sites (53.3%), intra-abdominal sites (36.7%), and in the abdominal wall (10%). Further, tumors recurred in 16 patients. To elucidate the relation between clinic-pathological characteristics and genetic alteration, we investigated the immunohistochemistry (IHC) for two proteins; alpha-smooth muscle actin(SMA) and *β*-catenin. These are also used for the diagnosis of desmoid. The focal positive expression of *β*-catenin was observed in 21 patients (70%), was not observed in five patients, and four patients were not evaluated. In IHC of alpha-SMA, expression was observed in 18 patients (60%), it was not observed in five patients, and seven patients were not evaluated.

We evaluated genomic alterations in the somatic mutation spectrum, somatic copy number alterations (SCNAs), and fusion transcripts in all 30 patients using target-seq (Figure 1A). The overall tumor mutation rate was low in desmoid tumors (1.76, range: 0–4.41) and similar to that in other sarcomas (Appendix A). The most frequent mutation was missense in CTNNB1 (=23, 77%, Figure 1B) at T41 and S45, with the following amino acid changes: T41A (=15, 65.2%), S45F (= 7, 30.4%), and S45P (= 1, 4.4%). The second most frequent mutation was in GNAQ (*n* = 18, 60%) at T95S and Y101. All patients with observed GNAQ mutation had missense at T95S (Figure 1B). Two patients (YD24 and YD25) with APC mutations were diagnosed with FAP.

All patients harbored at least one of these genetic mutations (CTNNB1, GNAQ and, APC). Figure 1C displays the relation between major genetic alteration and recurrency. Desmoid recurred in patients with only CTNNB1 mutation (*n* = 8, 80%), with only GNAQ mutation (*n* =1, 20%), and with CTNNB1 and GNAQ mutation (*n* = 5, 38.5%). Moreover, desmoid tumor recurred in all patients with APC mutation. From other research teams, information related to the prognostic role of CTNNB1 mutation was already reported. Especially, in meta-analysis, it was reported that the S45F mutation is a high-risk factor for the recurrence of desmoid [26].

### 3.3. Gene Expression Profiling

Based on transcriptomic data, we examined the signaling pathways that were enriched in desmoid tumors. Hierarchical clustering identified 1434 genes that were highly expressed in desmoid tumors compared to other sarcomas (Figure 2A), and gene ontology (GO) analysis revealed that these genes were significantly enriched in oncogenic pathways, such as the Hippo, TGF-*β*, and Wnt signaling pathways (Figure 2B). According to gene set enrichment analysis (GSEA), the top ten enriched pathways were the epithelial-mesenchymal transition, Wnt, pancreatic beta cells, estrogen response, TNF-*α*, Hedgehog, TGF-*β*, myogenesis, angiogenesis, and KRAS signaling pathways (Figure 2C,D).

Since the TGF-*β* signaling pathway was consistently identified as enriched in desmoid tumors, using both GO analysis and GSEA, we examined the cell type-specific expression profiles in response to TGF-*β* stimulation (also known as the TGF-*β* response signature; TBRS) in normal fibroblasts (F-), T cells (T-), macrophages (Ma-), and endothelial cells (End-) (27). Interestingly, the F-TBRS, T-TBRS, and Ma-TBRS scores were significantly enriched in desmoid tumors compared with other sarcomas, but the End-TBRS score was not (Figure 2E). A similar overall trend was observed in The Cancer Genome Atlas (TCGA) tumor dataset, in which desmoid tumors displayed relatively high F-TBRS and T-TBRS scores compared with other cancer types and sarcomas (Appendix A). Furthermore, samples with CTNNB1/APC and GNAQ mutations had significantly higher F-TBRS, T-TBRS, and Ma-TBRS scores, and lower End-TBRS scores (Figure 2F). These results suggest that the expression profiles of desmoid tumors are enriched in response to TGF-*β*, and that desmoid tumors may be dependent on the TGF-*β* pathway.

### 3.4. Patient-Derived primary Cell (PDC) as Preclinical Model Tools

We isolated and established PDCs from desmoid tumor tissue after surgical resection. Figure 3A represents the clinic-pathological information of desmoid tissues (YD12 and YD23) and microscopic image of PDCs (YD12-C and YD23-C). Two patient’s tissue presented the positive focal expression of *β*-catenin. The expression of alpha-SMA was present only in YD23. Moreover, desmoid tumor recurred in only YD23.

Next, we confirmed the PDCs by using target seq and RNA seq. Briefly, we described the genetic alteration information of three genes (CTNNB1, APC, and GNAQ) and expression score of TGFBR2, F-TBRS, and T-TBRS (Figure 3B). The detailed comparative table is presented in Appendix A. Each PDCs showed the same genetic alteration with their primary tissue. Moreover, the expression score of TGFBR2 and F-TBRS in YD12-C was similar with the primary desmoid tissue (YD12). In addition, YD23-C was at a similar level with the tissue (YD23). The TGFBR2 expression score was lower in YD23-C, and T-TBRS score was lower in both PDCs than their primary tissue.

In addition, the expression level of TGFBR2 was higher than F-TBRS and T-TBRS scores. Accordingly, based on our results, we suggested the combination treatment of vactosertib, a TGFBR type 1 inhibitor, for the enhancement of desmoid tumor suppression effect.

### 3.5. Desmoid Tumor Suppression Effect of Combination Treatment of TGFBR Type 1 Inhibitor

To conduct a comparison between combination treatment and monotherapy, we choose the imatinib monotherapy. Imatinib is a widely used drug for desmoid patients. Figure 3C shows the monotherapies and combination treatment. In YD12-C and YD23-C, imatinib and vactosertib monotherapy did not reach the half-maximal inhibitory concentration (IC_50_). However, it seems that imatinib monotherapy enhanced the growth of desmoid at low concentration in YD12-C. Moreover, vactosertib monotherapy did not exhibit the increment of tumor suppression, according to the rise of concentration. Therefore, we fixed the vactosertib concentration as 0.1 µM to clarify the effects of TGF-*β* pathway inhibition and desmoid tumor suppression. This concentration was selected by vactosertib monotherapy (Appendix A).

In combination treatment (Figure 3C), the viability of desmoid PDCs was lower than the monotherapy of imatinib and vactosertib. Prominently, the capability of tumor suppression was enhanced, even at low concentration of imatinib. Additionally, we stained the fluorescence by using live-dead viability/cytotoxicity kit (Figure 3D,E). The dead fluorescence signal was strongest in combination treatment with both desmoid PDCs. Conclusively, we confirmed that vactosertib combination improved the capability of tumor suppression.

### 3.6. Mechanism of Pharmacological Inhibition Using TGF-β Blockade

To confirm whether the observed suppression of the desmoid PDCs was due to the action of TGF-*β* inhibitors, we performed in-cell Western blot analysis to detect the signal of TGF-*β* path- way. Figure 4A presents an example fluorescence scanning image for in-cell Western blot analysis.

Figure 4B (YD12-C) and 4C (YD23-C) shows the measured fluorescence intensity of proteins and phosphorylation rate in desmoid PDCs. pTGFBR1 means the expression of phosphorylation in TGFBR1 at S165, and it is a signal of TGFBR dimerization, sending the signal to Smad2/3 proteins.

In YD12-C, all TGFBR2 expression levels with drug treatment were reduced. The reduction effect was higher in vactosertib monotherapy and combination treatment than imatinib monotherapy. TGFBR1 expression was highest in imatinib monotherapy (1.25) and lowest in combination treatment (0.73). The pTGFBR1 expression trend was similar with TGFBR1. It was lowest in combination treatment (0.69) and highest in imatinib monotherapy (1.17). In vactosertib monotherapy, the ratio of control of TGFBR1 and pTGFBR1 were similar with control (1.01, 0.99, respectively). In the Smad2/3 pathway, the trend of the ratio of control for Smad2/3 and pSmad2/3 expression in four sets of drug treatment was similar.

In YD12-C, TGFBR2 expression levels with drug treatment were reduced. Unlike YD12-C, the expression level of TGFBR1 was highest in vactosertib monotherapy and was lowest in imatinib monotherapy (There is no statistical significance). However, pTGFBR1 expression levels were decreased as follows: vactosertib (0.78), imatinib (0.66), and combination (0.52). In all of drug treatments, smad2/3 expression level was similar. However, pSmad2/3 was inhibited as follows: vactosertib (0.47), imatinib (0.73), and combination (0.63).

Subsequently, we normalized the protein expression considering the cell number and calculated the degree of phosphorylation (pTGFBR1/TGFBR1, pSmad2&3/Smad2&3). In each desmoid PDCs, the trend of the phosphorylation degree was similar between pTGFBR1/TGFBR1 and pS-mad2&3/Smad2&3. However, the effect of drug for receptor expression and phosphorylation was larger in YD23-C than YD12-C. A commonality in between PDCs was that combination treatment was more effective in terms of inhibition of the signal pathway contributing tumor growth.

Finally, confocal microscopy was used to confirm the detailed TGFBR1 and SMAD2/3 expression and phosphorylation in YD23-C (Figure 4D,E). In the control group, TGFBR1 expression and phosphorylation signals were present throughout in the cell. However, vactosertib clearly decreased TGFR1 expression and phosphorylation. SMAD2/3 expression and phosphorylation signals were stronger in the nucleus of the control cells but decreased following various drug treatments, particularly the combination treatment. Moreover, we measured the line profile of expression (Appendix A).

Based on the above results, it was confirmed that the tumor suppression effect was caused by a drug mechanism. Thus, we confirmed that the combination therapy of TGF-*β* inhibition and imatinib was more pharmacokinetically effective than individual treatments.

## 4. Discussion

In this study, using both molecular biology and bioinformatics methods, we showed that TGF-β receptor inhibitors could be effective clinical therapies for patients with desmoid tumors. To the best of our knowledge, this is the first study to demonstrate the potential efficacy of TGF-β inhibitors in desmoid tumors. In addition, we generated a patient-derived preclinical model from PDC lines that represented patient genomic alterations to demonstrate the proof-of-concept.

Previous genomic analyses of desmoid tumor profiles have reported the presence of CTNNB1 or APC mutations, but only in a few oncogenic lesions [27]. The current lack of a comprehensive molecular understanding of desmoid tumors has made it difficult for physicians to identify potential treatment options. Although a previous post-surgery predictive model provided a more precise comprehensive modality, its results could not be applied to treatment options [28]. Therefore, we conducted comprehensive molecular profiling using bioinformatic analyses to explore potential druggable targets.

We found that desmoid tumors harbor mutations in CTNNB1, APC, and GNAQ, all of which are associated with pathological activities [16]. For instance, alterations in the CTNNB1 and APC genes can lead to activation of the Wnt signaling pathway [29], whereas activating GNAQ mutations can lead to Hippo-independent activation of YAP pathways [30] and cancer progression. Previous studies have reported that the Wnt signaling pathway exhibits crosstalk with TGF-β signaling during fibroblast activations [31] and that YAP and TAZ can activate the TGF-β pathway via the transcription factors AP-1 and Smad7 in dermal fibroblasts [32]. Thus, both CTNNB1/APC and GNAQ mutations have been associated with increased TGF-β signaling in fibroblasts, and monoclonal proliferation can lead to desmoid tumors [33]. It is therefore unsurprising that we found that the expression of components of the TGF-β signaling pathway is enriched in desmoid tumors compared to that in other sarcomas and other cancers.

It has been suggested that TBRS, first reported by Calon et al. (2012) [25], reflects specific TGF-β pathway activation in major cell types in the tumor microenvironment and indicates the possibility of interaction with TGF-β [34,35]. In this study, the high TBRS scores in desmoid tumors compared to other sarcoma types may reflect their insensitivity to chemotherapy, and further suggest that targeting the TGF-β pathway could be an effective method for treating patients with desmoid tumors. Consistently, our transcriptomic data indicate that targeting TGF-β inhibition could be clinically relevant.

To validate this hypothesis, we established a patient-derived preclinical model and tested whether vactosertib could be an effective treatment regimen. However, vactosertib monotherapy appears to cause growth arrest rather than apoptotic cell death, yielding a similar response to γ-secretase inhibitors [36,37]. Therefore, we explored the potential of vactosertib in combination with imatinib, which is a standard treatment for desmoid tumors. This combination therapy was more effective than monotherapy, with imatinib requiring a high concentration, and the response to vactosertib varied on a case-by-case basis. Indeed, combination therapy effectively inhibited the growth of desmoid tumors in our patient-derived preclinical model, even at low concentrations. Consistently, our ongoing clinical trial, A Study to Evaluate the Safety and Efficacy of Vactosertib and Imatinib in Patients with Advanced Desmoid Tumor (ClinicalTrials.gov Identifier: NCT03802084), has also demonstrated clinical efficacy in desmoid tumors (Figure 5A,B). Together, these findings suggest that TGF-β inhibitors could be a useful treatment for desmoid tumors, and that this hypothesis should be confirmed using the final data of the clinical trial.

In this study, we also demonstrated that TGF-β inhibitors exert their anti-proliferative effects via both canonical (Smad) and non-canonical (non-Smad) signaling pathways. In particular, we found that low TGFBR1 expression reduced the dimerization ratio, which subsequently affected downstream components of the SMAD pathway. The combination treatment also markedly reduced TGFBR1 and SMAD2/3 protein phosphorylation, suggesting that the synergistic effect of the combination treatment originated from its molecular mechanism of action. The predicted mechanism of the interaction between nuclear β-catenin expression and the canonical TGF-β pathway is shown in Figure 5C. High-level nuclear β-catenin expression is a useful diagnostic marker for desmoid tumors with positive alpha-SMA staining [38,39]. CTNNB1 gene mutations are closely related to the degradation of CTNNB1. In patients with sporadic desmoid tumors, this degradation process does not proceed [40]. We confirmed that human β-catenin protein is encoded by the CTNNB1 gene (Appendix A) and is activated in the canonical TGF-β pathway by dimerization between TGF-βR2 and TGF-βR1. Downstream of the TGF-β pathway, SMAD3 and CTNNB1 form a complex with other molecules in the nucleus and induce EMT, proliferation, and angiogenesis in desmoid tumor cells via the SMAD2/3 pathway [41,42].

Although conventional commercially available cell lines are convenient and accessible, only a few are available for desmoid tumors. Therefore, it is important to establish preclinical models of desmoid tumors to allow the development of new drugs. In this study, we developed PDC models using surgically resected desmoid tumors that displayed the same morphological and genomic profiles. Previous studies have suggested that xenograft models derived from patients are better for cancer drug development, since PDC lines may reflect patient responses better than conventional cell lines. Although further studies are required to confirm the clinical efficacy of TGF-β inhibitors in patients with desmoid tumors, this pilot study and our preliminary clinical data suggest that this novel therapeutic strategy is promising.

## 5. Conclusions

In summary, this study demonstrated that fibroblasts, T-cells, and macrophages in desmoid tumors may have an increased response to TGF-β, and that the TGF-β signaling pathway may crosstalk with the Wnt and GNAQ pathways. Moreover, we confirmed the effect of combination treatment with the inhibitor of TGF-β pathway in our preclinical model. Together, these results suggest that TGF-β inhibitors could prove useful as a potential treatment for patients with desmoid, and that further research should be conducted to verify this.

## Figures and Tables

**Figure 1 cancers-14-05975-f001:**
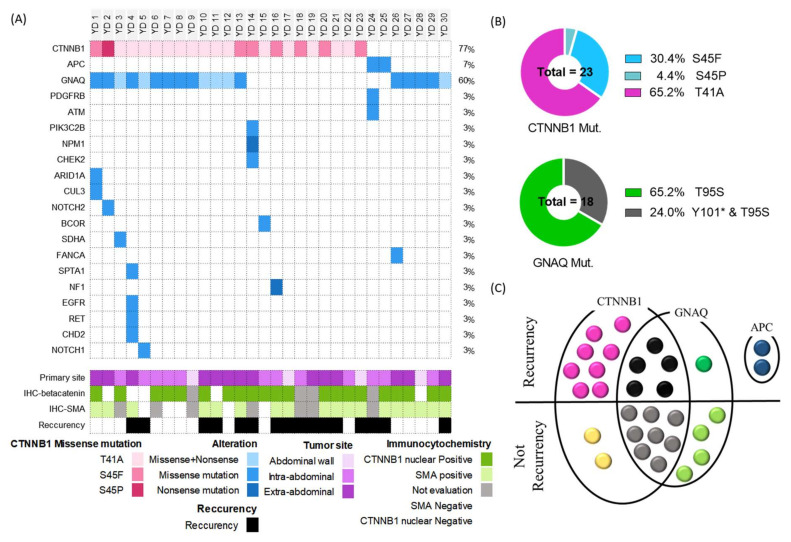
Genomic characteristics of desmoid tumors. (**A**) Landscape plot showing the alterations found by target-seq of 30 desmoid tumors. Each column represents a sample; each row represents a gene. Different colors indicate the type of genetic alteration, as shown at the bottom of the figure. (**B**) Pie charts for detailed mutation information of CTNNB1 and GNAQ. (**C**) Venn diagram displaying recurrence and major genetic alteration of the 30 cases patients. And, the dot represented the case.

**Figure 2 cancers-14-05975-f002:**
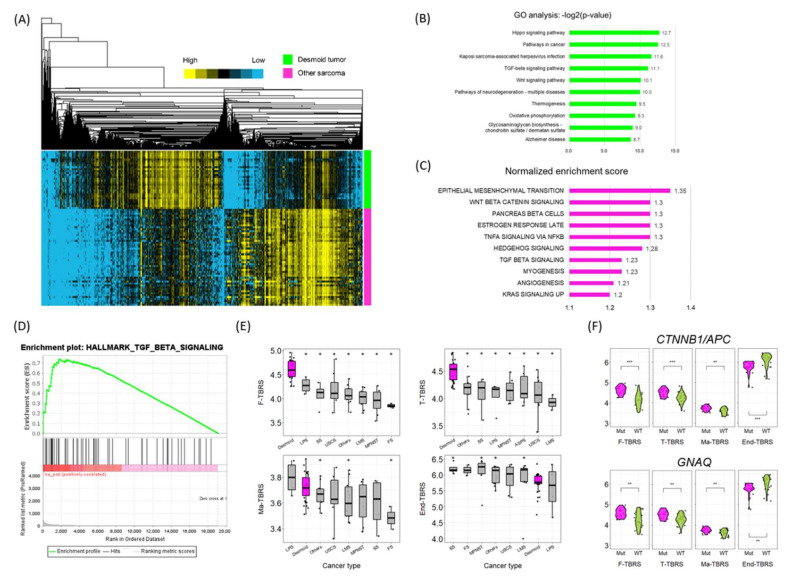
Gene expression profiles of desmoid tumors compared to other sarcomas. (**A**) Heatmap showing the results of hierarchical clustering. Rows represent patients; columns represent genes. The type of tumor is annotated at the right of the heatmap: green indicates desmoid tumors; magenta indicates other sarcomas. Expression levels are indicated in different colors, with yellow being high and blue being low. (**B**) Bar graph showing the results of GO and Kyoto encyclopedia of genes and genomes pathway analyses. (**C**) Bar graph showing the results of GSEA. Only the top ten pathways with nominal *p* values < 0.05 are shown. (**D**) GSEA plot of the TGF-β signaling pathway. (**E**) Boxplots showing F-TBRS, T-TBRS, Ma-TBRS and End-TBRS scores according to cancer type. Magenta boxes represent desmoid tumors; grey boxes represent other sarcomas. Cancer types with significantly different values (*p* < 0.05) compared with desmoid tumor are marked with asterisks (*).Each dot in the plots represent the value of each sample. (**F**) Violin plots showing TBRS scores according to CTNNB1/APC (*n* = 23 for Mut and *n* = 24 for WT) or GNAQ (*n* = 18 for Mut and *n* = 29 for WT) mutation status. Each dot in the plots represent the value of each sample. *** *p* < 0.001, ** *p* < 0.01, * *p* < 0.05; Wilcoxon rank-sum test. Abbreviations: FS, fibrous sarcoma; LMS, leiomyosarcoma; LPS, liposarcoma; MPNST, malignant peripheral nerve sheath tumor; Mut, mutated; SS, synovial sarcoma; USCS, undifferentiated spindle cell sarcoma; WT, wild type.

**Figure 3 cancers-14-05975-f003:**
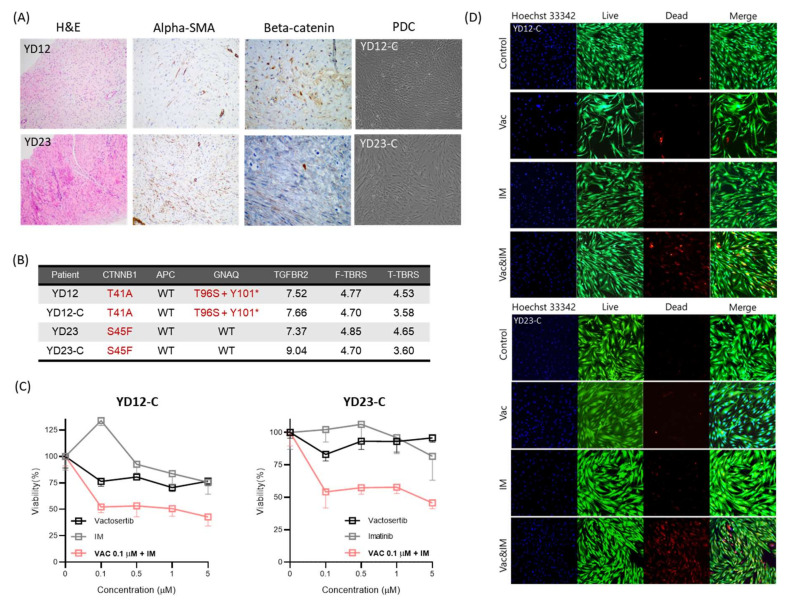
Preclinical model characteristics and anti-drug response. (**A**) H &E, SMA, and beta-catenin staining image of desmoid tissue (YD12 and YD23) and microscopic image of patient-derived preclinical models (YD12-C and YD23-C). (**B**) Most prevalent genetic alterations and TGFBR2, F-TBRS, and T-TBRS scores. YD12 had genetic alterations in CTNNB1 and GNAQ, while YD23 had CTNNB1 mutations. TGFBR2 and F-TBRS RNA expression were similar between YD12 and YD12-C. TGFBR2 expression was higher in YD23-C cells than in tumor tissue. F-TBRS levels were similar in YD23-C and YD23. T-TBRS levels were lower in both cell lines. (**C**) Cell viability with monotherapy (Vac; vactosertib, IM; imatinib) and combination treatment (Vactosertib (0.1 μM) and imatinib) in the patient-derived preclinical models. (**D**) Fluorescence microscopy image of the live-dead assay in the patient-derived preclinical models.

**Figure 4 cancers-14-05975-f004:**
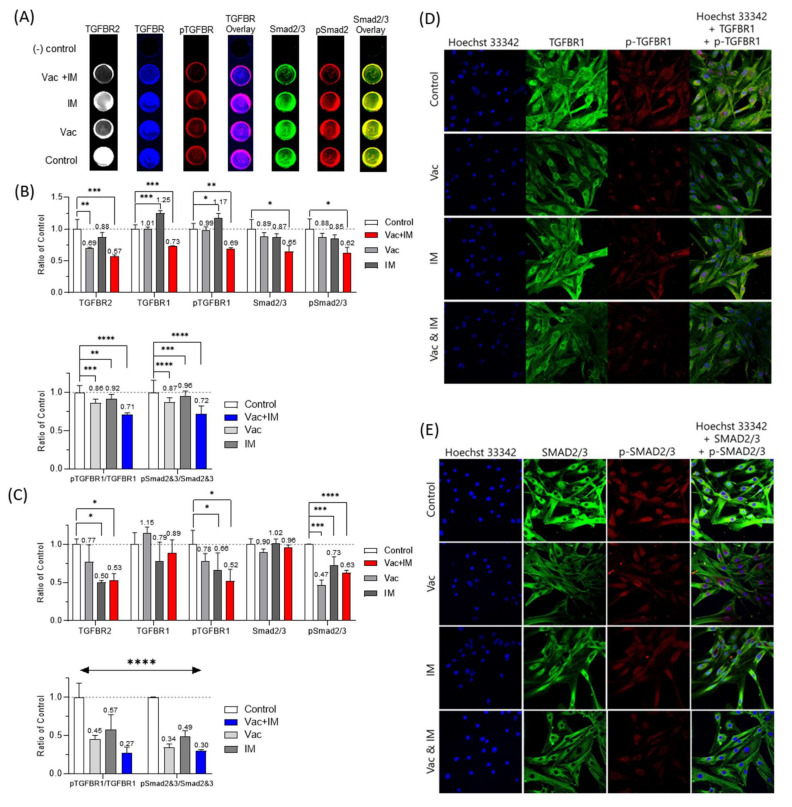
In-cell western blot analysis. (**A**) Example of fluorescence scanning image with in-cell western blot (YD12-C). All images on top was (-) control. The ratio of control gar graph for protein and phosphorated protein and the degree of phosphorylation in YD12-C (**B**) and YD23-C (**C**). (**D**,**E**) were confocal microscopic fluorescence image of YD23-C with drug treatment. All drug concentration was 0.1 µM. The line profile of fluorescence intensity was in Appendix A. * *p* < 0.05, ** *p* < 0.01, *** *p* < 0.001, and **** *p* < 0.0001.

**Figure 5 cancers-14-05975-f005:**
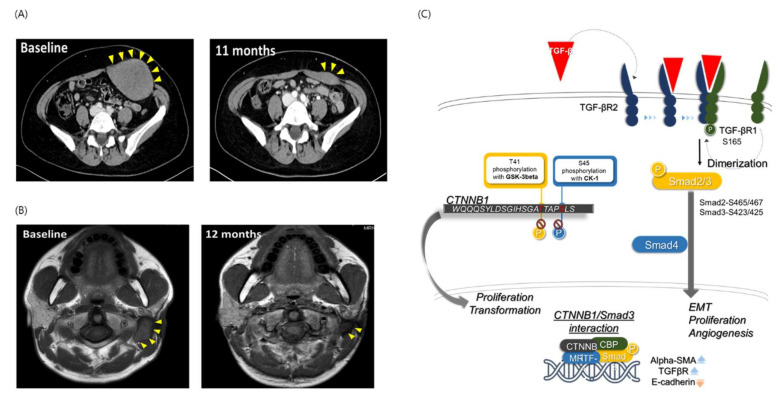
CT images of patients with desmoid tumors and schematic illustration of the interactions between the clinical and genomic properties of desmoid tumors. CT images from the ongoing clinical trial for the combination treatment (**A**,**B**). (**C**) Interaction between the clinical (beta-catenin and SMA expression) and genomic (CTNNB1 mutation and TGF-β signaling) properties of desmoid tumors.

## Data Availability

Data are available in the publications cited in the manuscript. Further information and requests for resources and reagents should be directed to and will be fulfilled by lead contact with Hyo Song Kim.

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
