# Peer review of "Therapeutic Implications of TGF-β Pathway in Desmoid Tumor Based on Comprehensive Molecular Profiling and Clinicopathological Properties"

_cancers, 2022, doi:10.3390/cancers14235975_

Round 1

Reviewer 1 Report

My comments (also attached in word file):

First of all I think it would be appreciated a language revision to optimize some paragraphs.

Then:

1.       From line 50

1 Introduction: please specify that DTs are locally aggressive but non-metastasizing tumors. I suggest to mention the “watch and wait” strategy in the management of DTs given the possibility of stabilization and spontaneous regression in some cases.

So please add recent  literature such us Kasper B et Al Ann Oncol 2017 (DOI: 10.1093/annonc/mdx323)

2.       From line 94

2.2   Tumor sample collection: please specify the reason why you chose a cut off of 40% for the tumor content instead of  a measure in square millimetre (mm2).

From line 182

2.6 Immunohistochemistry: this paragraphs is quite the same of a previous work published by your 

      team. It would be appreciated a reference to that paper.

From line 213

2.9 In-cell western blotting: what kind of drugs are you referring to?

3.       Line 223, line 232: “had the chemotherapy history”; “For fine out”

These are some example to explain the necessity of language revision

From Line 249: It could be useful to mention literature data about the prognostic role of CTNNB1 mutation.

Please cite the phase I/II studies evaluating the combination of vactosertib and imatinib in patients with DT. Note that the phase II study has been withdrawn (not for efficacy and safety of vactosertib but changes in the company development strategies). Obviously this information is not necessary for the purpose of the present work.

Last question:

-          It would have been useful to evaluate the false discovery rate (FDR), often used to correct for multiple comparisons to strengthen the p-value. Could you explain why you didn’t use it?

Reviewer 2 Report

In this paper, Yun et al. investigated the molecular significance of TGFb signaling in desmoid tumors and suggested an efficient combination therapy approach to treat these tumors.

Below you can find my major and minor comments about this paper. 

Major comments

1. An isobologram analysis can be helpful to show a clear drug-drug interaction response. By doing this experiment, you will be able to calculate a CI, which shows synergistic, antagonistic or additive effect between two suggested drugs in the manuscript. It will also be helpful to determine real IC50 values of individual drugs and the combination.

 Minor comments

1. Individual data points should be shown on the bar graphs.

2. Figure 5c (graphical schematic) can be improved with bigger labels. 

Author Response

Thank you.

Round 2

Reviewer 1 Report

The only comment relate to the following paragraph:

Nowadays, many patients are taking the method of waiting carefully after diagnosis. Because , sometimes, DTs given the possibility of stabilization and spontaneous regression [1] . hen the desmoid 53 start to grow, major treatment for primary care is surgery [2,3] 

The watchful waiting approach is part of the therapeutic strategy not a patient decision 

Reviewer 2 Report

Dear Authors, 

Thanks for your response to the comments in the first version of your paper. I do not know if it is a technical issue or not but I could not see individual data points on your bar graphs. Please ignore this note if it is a technical error, but I would like to remind you to change graphing options for your bar graphs. Congratulations on your new publication!
